# UK poSt Arthroplasty Follow-up rEcommendations (UK SAFE): what does analysis of linked, routinely collected national datasets tell us about mid–late term revision risk after knee replacement?

Lindsay K Smith [1], Cesar Garriga,[2] Sarah R Kingsbury,[3,4] Rafael Pinedo-Villanueva,[2] Antonella Delmestri,[2] Nigel K Arden,[5,6] Martin Stone,[7] Philip G Conaghan [4], Andrew Judge[8]

► http://dx.doi.org/10.1136/bmjopen-2021-050877

**Correspondence to**
Dr Lindsay K Smith;
lindsay.smith2@nhs.net

## ABSTRACT

**Objective** To identify patients at risk of mid-late term revision of knee replacement (KR) to inform targeted follow-up.

**Design** Analysis of linked national datasets from primary and secondary care (Clinical Practice Research Datalink (CPRD GOLD), National Joint Registry (NJR), English Hospital Episode Statistics (HES) and Patient Reported Outcome Measures (PROMs)).

**Participants** Primary elective KRs aged ≥18 years.

**Event of interest** Revision surgery ≥5 years (mid–late term) postprimary KR.

**Statistical methods** Cox regression modelling to ascertain risk factors of mid–late term revision. HRs and 95% CIs assessed association of sociodemographic factors, comorbidities, medication, surgical variables and PROMs with mid–late term revision.

**Results** NJR-HES-PROMs data were available from 2008 to 2011 on 188 509 KR. CPRD GOLD-HES data covered 1995–2011 on 17 378 KR. Patients had minimum 5 years postprimary surgery to end 2016. Age and gender distribution were similar across datasets; mean age 70 years, 57% female. In NJR, there were 8607 (4.6%) revisions, median time-to-revision postprimary surgery 1.8 years (range 0–8.8), with 1055 (0.6%) mid–late term revisions; in CPRD GOLD, 877 (5.1%) revisions, median time-to-revision 4.2 years (range 0.02–18.3), with 352 (2.0%) mid–late term revisions.

Reduced risk of revision after 5 years was associated with older age (HR: 0.95; 95% CI 0.95 to 0.96), obesity (0.70; 0.56 to 0.88), living in deprived areas (0.71; 0.58 to 0.87), non-white ethnicity (0.58; 0.43 to 0.78), better preoperative pain and functional limitation (0.42; 0.33 to 0.53), better 6-month postoperative pain and function (0.33; 0.26 to 0.41) or moderate anxiety/depression (0.73; 0.63 to 0.83) at primary surgery.

Increased risk was associated with male gender (1.32; 1.04 to 1.67); when anticonvulsants (gabapentin and pregabalin) (1.58; 1.01 to 2.47) or opioids (1.36; 1.08 to 1.71) were required prior to primary surgery.

> **Strengths and limitations of this study**
>
> ► This study is part of a wider programme of work to identify potential patient groups for follow-up after hip and knee replacement and used large national routine datasets from primary and secondary care.
> ► The linkage of datasets allowed us to explore the impact of multiple risk factors on the mid–late term risk of revision of knee replacement.
> ► This is one of the first studies to identify predictors of mid–late term revision risk for knee replacement from real-world data and contributes to the discussion on follow-up.
> ► A limitation of the National Joint Registry–Hospital Episode Statistics–Patient Reported Outcome Measures linked data was limited long-term follow-up due to including data from 2009 onwards but only primary operations up to 2011 to allow for revision rates after 5 years.

No implant factors were identified.

**Conclusion** The risk of mid–late term KR revision is very low. Increased risk of revision is associated with patient case-mix factors, and there is evidence of sociodemographic inequality.

## INTRODUCTION

Primary knee replacement (KR) surgery is a common elective orthopaedic procedure for the treatment of knee pain due to end stage osteoarthritis (OA). There is good evidence showing that KR is highly clinically effective, reducing symptoms of pain and functional limitations for the vast majority of patients[1–3] and is also cost effective.[4 5] Over 100 000 operations are carried out each year in the UK.[6] The lifetime risk of receiving



knee arthroplasty in the UK is estimated to be 10.8% for women and 8.1% for men.[7] These numbers are projected to increase with an ageing and increasingly obese population, placing a growing public health burden on the National Health Service (NHS) in respect of funding and capacity.[8]

There is significant pressure on hospital trusts to reduce the amount of follow-up appointments due to expanding waiting lists, cancellation of elective surgery and increasing numbers of patients needing primary joint replacement. Although previous British Orthopaedic Association guidelines recommended outpatient follow-up at 1 and 7 years, and every 3 years thereafter, recent guidelines for primary joint replacement in the UK recommend further research on follow-up due to a lack of evidence.[9 10] There is variation across the country in how hospitals organise follow-up services, and many units stopped follow-up after an early postoperative check.[11] Evidence is required on the impact that disinvestment in follow-up services may have on patient safety. There is a need to ensure early detection of patients with failing implants and target follow-up accordingly. In March 2014, the James Lind Alliance and National Institute for Health Research (NIHR) Priority Setting Partnership for Hip and Knee Replacement for Osteoarthritis identified that defining the ideal postoperative follow-up period and the best long-term care model for people with OA and knee replacement was among its top 10 research priorities, highlighting the importance of appropriate follow-up to ensure the health of patients.

The objective of this study was to use nationally available datasets to identify which groups of patients with KR may require follow-up based on their mid–late term revision risk (five or more years post primary surgery). This work forms part of a larger programme of work, UK SAFE, that was designed to address the research question: is it safe to disinvest in mid–late term follow-up of hip and knee replacement?[12] The UK SAFE programme of work took place between 1 December 2016 and 30 November 2020 (protocol provided in online supplemental file 1).

## METHODS
### Study design
This was a nationwide retrospective cohort study in which national data from primary care (Clinical Practice Research Datalink) and secondary care (National Joint Registry (NJR), Hospital Episode Statistics (HES) and Patient-Reported Outcome Measures) were linked to identify predictors of mid–late term revision of KR.

### Sources of data
#### Clinical Practice Research Datalink (CPRD)-GOLD-HES
The CPRD GOLD comprises the entire computerised medical records of a sample of patients attending general practitioners (GPs) in the UK.[13] It contains information on over 14 million patients registered at over 700 general practices in the UK. With 4.4 million active (alive,

currently registered) patients meeting quality criteria, approximately 6.9% of the UK population are included, and patients are broadly representative of the UK general population in terms of age, sex and ethnicity.[14] GPs in the UK play a key role in the delivery of healthcare by providing primary care and referral to specialist hospital services, and each GP practice records this medical information for individual patients. The CPRD is administered by the Medicines and Healthcare products Regulatory Agency. CPRD GOLD records contain all clinical and referral events in both primary and secondary care in addition to comprehensive demographic information, prescription data and hospital admissions. Data are stored using Read codes for diseases that are cross-referenced to the International Classification of Diseases (ICD-10). Read codes are used as the standard clinical terminology system within UK primary care. Only practices that pass quality control are used as part of CPRD GOLD. CPRD ensures patient confidentiality by providing anonymised healthcare records.

CPRD GOLD data were linked to data for all-cause mortality, provided by the Office for National Statistics.[15] CPRD GOLD data were also linked to the Index of Multiple Deprivation (IMD) and to the HES database (described later). CPRD already provide access to HES data for England that is held under the CPRD data Linkage Scheme, available for around 60% of patients in the CPRD GOLD database. Previous research by the CPRD team has shown that linked practices/patients are representative of the CPRD GOLD population as a whole.[16]

### NJR–HES–Patient Reported Outcome Measures (PROMs)
Starting in 2003, the National Joint Registry (NJR) collected information on all hip and knee replacements performed each year in both public and private hospitals in England, Wales, Northern Ireland and the Isle of Man.[17] Data are entered into the NJR using forms completed at the time of surgery, and revision operations are linked to primaries using unique patient identifiers. Data recorded in the NJR includes prosthesis and operative information (prosthesis type, approach and thromboprophylaxis); patient information (age, gender, body mass index (BMI), American Society of Anaesthesiologists (ASA) grade); and surgeon and unit information (including caseloads and public/private status).

The HES database holds information on all patients admitted to National Health Service (NHS) hospitals in England, including diagnostic ICD codes providing information about a patient's illness or condition and NHS national clinical procedural codes (OPCS4) for surgery. It covers a smaller geographical area than the NJR and does not include privately funded operations. However, HES provides additional information for every patient (including detailed comorbidity information and deprivation indices) and about every procedure (including length of stay and need for blood transfusion or critical care). Additional records contain details of readmissions,

reoperations and revisions not recorded in the NJR database.

Since April 2009, PROMs data have been collected on all knee replacements performed in public hospitals in England.[18] A health-related quality of life questionnaire (the EuroQol with five domains (EQ-5D-3L)[19]) and a joint-specific outcome score (the Oxford Knee Score (OKS)[20]) are collected preoperatively and at 6 months after surgery, along with patient-reported measures of preoperative disability and postoperative satisfaction.

For this analysis, we used NJR records linked to data from the HES and PROMs databases on all KR operations.

## Participants

Anonymised records were extracted for all patients over 18 years of age receiving primary knee replacement surgery. For CPRD GOLD-HES data, the time span covered the years 1995–2017; for NJR–HES–PROMS data, it covered the years 2009–2017. Patients were included if they had primary total knee replacement or unicompartmental knee replacement. We excluded patients that had revision surgery and total joint replacement of unspecified fixation. The following exclusions were made to remove potential case-mix issues: other injuries due to trauma, such as transport accidents and falls; non-elective admissions; and a diagnosis other than primary knee OA. There will be some overlap between patients receiving knee replacement in the two data sources (around 7% of patients between 2009 and 2016); however, these anonymised datasets are analysed independently of each other.

## Primary outcome

Early complications (defined as less than 5 years) are often symptomatic and include infection and technical errors.[21] Arthroplasty failure in the longer term (defined as after 5 years), constituting 50% of revision surgery, is usually caused by bearing-surface wear and associated consequences of periprosthetic osteolysis or aseptic loosening and may be asymptomatic until clinical and radiographic failure have occurred.[21 22] The primary outcome was defined as mid–late term revision (defined as more than 5 years postprimary surgery). Revision is defined as the removal, exchange or addition of any of the components of arthroplasty. In the NJR–HES–PROMS linked datasets, operative details are completed using the NJR dataset, rather than the OPCS4 coding used by the HES dataset. The NJR collects operative data using two forms: one for primary operations and the other for revision operations. In both cases, all component labels from the surgery are attached to the form, and it is from these that the component details are collected. Revision operations are linked to primaries using unique patient identifiers and so, two operations on the same knee would be linked using this system. The combination of the separate coding at source and the secondary linkage gives confidence that primary and revision operations are correctly identified. In the CPRD GOLD dataset, subjects with a revision surgery procedure are identified using the Read codes, and for those with HES-linked data OPCS4 codes can be used.

## Predictors

### Secondary care predictors

The patient level characteristics available in NJR and HES include: age, gender, BMI, area deprivation, rurality, ethnicity, Charlson comorbidity index[23] (calculated from HES using ICD10 codes), ASA grade. Data from the NJR provide additional information on surgical and operative factors: whether or not a minimally invasive technique was used; annual surgeon volume/case load, operative time, grade of operating surgeon, surgical approach, patient position, implant fixation, type of mechanical or chemical thromboprophylaxis and unit type (public, private, independent sector treatment centre). Data from the PROMs database provide additional information on symptoms of pain, function and health related quality of life preoperatively and at 6 months postsurgery. Pain and function are measured using the OKS. The EQ-5D-3L consists of five questions (assessing mobility, self-care, ability to conduct usual activities, degree of pain/discomfort and degree of anxiety/depression), ranging from 1 (best state) to 3 (worst state). EQ-5D-3L can be expressed as an overall index (graded from −0.594 to 1), or as ordinal responses for each category.

### Primary care predictors

The CPRD GOLD database includes information on: age, gender, BMI, joint replaced (hip/knee), year of joint replacement operation, recorded diagnosis of OA (yes/no), fracture presurgery (yes/no), calcium and vitamin D supplements, use of bisphosphonates, use of selective oestrogen receptor modulators, oral glucocorticosteroid therapy, smoking status and alcohol intake recorded closest to the date of the primary surgery, region of UK, comorbid conditions registered by the physician from the following list (asthma, malabsorptive syndromes, inflammatory bowel disease, hypertension, hyperlipidaemia, ischaemic heart disease, stroke, chronic obstructive pulmonary disease, chronic kidney failure, neoplasms, diabetes), use of drugs that can affect fracture risk (proton pump inhibitors, antiarrhythmics, anticonvulsants, antidepressants, anti-Parkinson drugs, statins, thiazide diuretics and anxiolytics).

## Sample size

We included all patients receiving planned elective primary surgery for knee OA. For the NJR–HES–PROMs data, this covered the years 2009–2016 (as our requested linked HES data was from 2008 onwards, and earlier years of data were not available to us). For the CPRD GOLD-HES, this spanned the years 1995–2016. For both datasets, we excluded patients receiving a primary knee replacement after 2011 to ensure all patients had at least 5-year follow-up, as we were not interested in revisions occurring in the early period up to 5 years after the

primary replacement surgery. The sample was created from all available data that satisfied these criteria.

## Statistical analysis methods

Survival analysis was used to model time to revision. To identify patients most likely to require revision, proportional hazards regression modelling was used to identify preoperative, perioperative and postoperative predictors of mid–late term revision. The date of the first incidence of a subject's knee replacement was used as the start time. The event of interest in all time-to-event models was the first recorded revision operation. Linearity of continuous predictors was assessed using fractional polynomial regression modelling. Proportionality assumptions were checked using Schoenfeld residuals. Missing data were handled by using multiple imputation methods using the Imputation by Chained Equations procedure.[24] SEs were calculated using Rubin's Rules. We include all predictor variables in the multiple imputation process, together with the outcome variable (Nelson Aalen estimate of survival time and whether or not the patient had the outcome) as this carried information about missing values of the predictors.

For the CPRD GOLD-HES primary care, we generated 10 imputed datasets for KR. Data were imputed for the variables BMI, deprivation index, smoking and drinking risk factors. For secondary care NJR–HES–PROMS dataset, we generated a single imputed dataset for KR. Variables imputed were BMI, deprivation index, rurality, ethnicity, OKS baseline scores and EQ-5D-3L item for anxiety and depression. We ran univariate Cox regression models. Risk factors with a p value <0.20 were selected for a multivariable model. Backward selection of variables was used to identify variables to keep in the final model risk factors with at least one category with a p value <0.05. For the CPRD GOLD-HES primary care dataset, we present two final models: one with medication use as yes/no variables and the other model with daily defined doses (DDDs) calculated from 1 year prior to the primary surgery and divided in tertiles. In addition, we conducted sensitivity analyses using a Fine-Gray competing risk model to account for the competing risk of death.

## Patient and public involvement

Members of the NIHR Leeds Biomedical Research Centre and Bristol public and patient involvement groups (PPI) were involved in developing the UK SAFE research question and work programme based on experiences of arthroplasty and preferences for care. The steering committee includes a PPI coapplicant who has contributed to interpretation of the results and will be involved in production of the final report that is disseminated to the public, patients and NHS staff.

## RESULTS

This study has been reported in accordance with the Strengthening the Reporting of Observational Studies in Epidemiology checklist (online supplemental file 2).

**Table 1** Stages of patient selection for inclusion in study: primary care data

| Included | Excluded |
|---|---|
| Patients with primary knee replacement in CPRD GOLD (64 071) | |
| → | Outcome (knee revision) and index event (primary surgery) outside England: 5397 (8.4%); Wales 6982 (10.9%) |
| → | Outcome (knee revision) and index event: 31 395 (42.9%) |
| Patients with primary knee replacement in CPRD GOLD linked to HES and used in survival analysis (22 836) | |
| → | Primary surgery after 2011 (allowing for 5 years of follow-up): 5458 (23.9%) |
| Patient with primary knee replacement used in the survival analysis (17 378) | |

CPRD, Clinical Practice Research Datalink; HES, Hospital Episode Statistics.

## Participants

For the CPRD GOLD-HES dataset, 64 071 sets of data were available, and table 1 shows the steps towards 17 378 participants. Construction of the NJR–HES–PROMs dataset commenced with 84 1212 records in the NJR and 188 509 participants after exclusions (table 2).

Summary statistics for patients in the CPRD GOLD-HES and the NJR-HES-PROMs linked datasets are provided (online supplemental file 3, tables A,B. The CPRD GOLD-HES linked data covered a longer time period between 1995 to 2011; the NJR-HES-PROMs data were available 2009–2011. The characteristics of patients in the full CPRD dataset compared with those in the CPRD-HES linked data were similar with no evidence of any selection bias (online supplemental file 3, table C). Both datasets allowed a minimum of 5-year follow-up to end 2016. The age and gender distribution of patients was similar across both datasets, with a mean age of 70 years at time of knee replacement and 57% female. An extensive range of patient case-mix, surgical, operative factors and primary care prescribing data was available for analysis.

The CPRD GOLD-HES dataset had a longer time to revision. There were 877 (5.1%) revisions, with median time to revision of 4.2 years (range 0.02–18.3 years) and 352 (2.0%) were mid–late term revisions.

In the NJR-HES-PROMs data, there were 8607 (4.6%) knee replacement revisions with a median time to revision of 1.8 years (range 0–8.8 years); this included 1055 (0.6%) mid–late term revisions.

**Table 2** Stages of patient selection for inclusion in study: hospital data

| Included | | Excluded |
|---|---|---|
| Patients with primary knee replacement in National Joint Registry (841 212) | | |
| | → | Primary surgery before 2008 (no data available in HES) (169 776; 20.2%) |
| | → | Primary surgery after 2011 (allowing for 5 years of follow-up) (414 832; 49.4%) |
| | → | Without primary surgery date (1037; 0.1%) |
| | → | A diagnosis other than primary knee osteoarthritis (2940; 0.4%) |
| | → | Non-elective surgeries (535; 0.06%) |
| | → | Without information on type of admission (63 416; 7.5%) |
| Patient with primary knee replacement used in the survival analysis (188 509) | | |

HES, Hospital Episode Statistics.

## Predictors of mid–late term revision

### Patient demographics

Older age at the time of primary KR was associated with a lower risk of mid–late revision (tables 3 and 4). The effect of age was linear and the association was strong where, for a 1-year increase in age at surgery, the risk of outcome reduced by 5%, and this finding was consistent across the CPRD GOLD-HES and NJR–HES–PROMs datasets. The effect of gender was that males had an increased risk of mid–late revision compared with females. This was only observed in the CPRD GOLD-HES data, where males had a 24% increased risk of revision, while the effect size was weaker and non-significant in the NJR-HES-PROMs dataset.

The effect of obesity on outcome was demonstrated in the NJR–HES–PROMs dataset, where compared with those of normal BMI, underweight patients were at increased risk of revision and obese patients at reduced risk of mid–late revision. The effect of IMD deprivation in the NJR–HES–PROMs dataset showed that patients in the most deprived areas were less likely to receive mid–late term revision; there was no such association with obesity or deprivation observed in the CPRD GOLD-HES dataset.

An association with ethnicity was observed only in the NJR–HES–PROMs dataset, where patients of non-white ethnicity were less likely to be revised mid–late term.

### Implant factors (NJR–HES–PROMs dataset)

None of the implant related factors were associated with an increased mid–late revision risk.

### Preoperative and 6-month follow-up PROMs (table 4)

There was a clear linear trend with the preoperative and 6-month postoperative OKS, where patients with the most pain and functional limitations at the time of surgery, and at 6 months after surgery, were substantially more likely to require mid–late term revision. Patients with preoperative anxiety/depression were found to be less likely to receive a mid–late revision operation.

### Primary care comorbidities and medication use (table 3)

Through the CPRD GOLD-HES dataset, we were able to investigate comorbidities recorded prior to surgery and medication use. There was no effect of preoperative comorbidity for KR. With medication use, oral glucocorticoid steroid therapy was associated with a lower risk of revision, whereas use of antiarrhythmics and anticonvulsants placed patients at a higher risk.

For the pain medication use, an increased revision risk was observed in those patients requiring opiates. When examining effects of medication use in more detail, by looking at DDDs calculated from the 1 year prior to the primary surgery and divided into tertiles, the effect of opioids was only significant in the highest DDD tertile of >600 DDD.

The sensitivity analysis for the competing risk of death is presented (tables 3 and 4).

## DISCUSSION

The risk of a mid–late revision operation (≥5 years) after primary knee replacement surgery is very low. Within our CPRD GOLD-HES primary care dataset, we had up to 20 years patient follow-up from the start point of 5 years after the primary operation and even then, the mid–late revision rate was only 2.0%. In this study, it was the patient case-mix factors that were associated with mid–late term revision surgery. Patients at increased risk were those who were younger, male gender, not obese, living in affluent areas, of white ethnicity, not anxious or depressed at primary surgery. Those with worse pain and functional scores at primary surgery were at higher risk for mid–late revision than those with better scores.

Strengths of this study include the use of large national routine datasets where the NJR data are mandatory and have near complete coverage, and the CPRD GOLD data is nationally representative in respect of UK population demographic characteristics. Large sample sizes afforded us the ability to identify predictors of a rare long-term outcome such as revision surgery. A limitation of the NJR–HES–PROMs linked data was limited long-term follow-up

**Table 3** Cox regression model identifying risk factors for revision after 5 years of primary total knee and unicompartmental replacement: primary care data

| Risk factors (reference category) | Patients undergoing KR (n=17 378) | | | Patients undergoing KR with missing dose for bisphosphonates and opioids excluded (n=14 470) | |
| --- | --- | --- | --- | --- | --- |
| | Crude analysis | Adjusted analysis | Adjusted competing risk analysis | Adjusted analysis | Adjusted competing risk analysis |
| | | (Drug yes/no) | (Drug yes/no) | (Drug DDD) | (Drug DDD) |
| | HR (95% CI) | HR (95% CI) | HR (95% CI) | HR (95% CI) | HR (95% CI) |
| | P value | P value | P value | P value | P value |
| **Year of primary KR (2010–2011)** | | | | | |
| 1995–1999 | 4.63 (1.98 to 10.81); **p<0.01** | 5.39 (2.28 to 12.75); **p<0.01** | 6.60 (2.82 to 15.44); p<0.01 | 8.10 (2.52 to 25.98); **p<0.01** | 10.16 (3.20 to 32.29); p<0.01 |
| 2000–2004 | 3.24 (1.42 to 7.41); **p=0.01** | 3.65 (1.59 to 8.40); **p<0.01** | 4.33 (1.90 to 9.87); p<0.01 | 5.49 (1.73 to 17.37); **p<0.01** | 6.64 (2.12 to 20.83); p=0.001 |
| 2005–2009 | 2.36 (1.04 to 5.36); **p=0.04** | 2.42 (1.06 to 5.52); **p=0.04** | 2.77 (1.22 to 6.28); p=0.015 | 3.45 (1.10 to 10.86); **p=0.03** | 4.04 (1.29 to 12.65); p=0.017 |
| Age at primary KR (continuous variable) | 0.93 (0.92 to 0.94); **p<0.01** | 0.93 (0.92 to 0.94); **p<0.01** | 0.93 (0.92 to 0.93] ; p<0.01 | 0.93 (0.92 to 0.94); **p<0.01** | 0.92 (0.92 to 0.93); p<0.01 |
| **Sex (woman)** | | | | | |
| Man | 1.26 (1.02 to 1.55); **p=0.03** | 1.24 (1.00 to 1.53); p=0.06 | 1.18 (0.95 to 1.46); p=0.13 | 1.32 (1.04 to 1.67); p=**0.02** | 1.26 (1.00 to 1.60); p=0.054 |
| **Body mass index (normal)** | | | | | |
| Underweight | | | | | |
| Overweight | 1.02 (0.71 to 1.45); p=0.93 | 0.97 (0.67 to 1.42); p=0.89 | 1.01 (0.69 to 1.47); p=0.96 | 0.98 (0.65 to 1.46); p=0.91 | 1.01 (0.68 to 1.51); p=0.97 |
| Obese class I (moderately obese) | 1.25 (0.86 to 1.80); p=0.24 | 1.06 (0.71 to 1.57); p=0.79 | 1.08 (0.73 to 1.60); p=0.71 | 1.09 (0.69 to 1.70); p=0.72 | 1.11 (0.71 to 1.73); p=0.66 |
| Obese class II and higher | 1.35 (0.91 to 2.00); p=0.14 | 1.03 (0.65 to 1.63); p=0.90 | 1.03 (0.65 to 1.64); p=0.90 | 0.97 (0.58 to 1.63); p=0.90 | 0.97 (0.57 to 1.63); p=0.89 |
| **Region (East Midlands)** | | | | | |
| East of England | 0.83 (0.49 to 1.41); p=0.49 | 0.95 (0.56 to 1.61); p=0.84 | 0.94 (0.55 to 1.59); p=0.82 | | |
| London | 0.81 (0.46 to 1.43); p=0.47 | 0.96 (0.54 to 1.71); p=0.90 | 0.94 (0.53 to 1.66); p=0.83 | | |
| North East | 0.28 (0.08 to 0.95); **p=0.04** | 0.27 (0.08 to 0.91); **p=0.04** | 0.27 (0.08 to 0.91); p=0.035 | | |
| North West | 0.88 (0.53 to 1.47); p=0.63 | 0.93 (0.56 to 1.55); p=0.78 | 0.91 (0.55 to 1.52); p=0.73 | | |
| South Central | 0.81 (0.48 to 1.36); p=0.42 | 0.93 (0.55 to 1.57); p=0.79 | 0.91 (0.54 to 1.52); p=0.71 | | |
| South East Coast | 1.08 (0.64 to 1.82); p=0.77 | 1.37 (0.82 to 2.29); p=0.23 | 1.33 (0.80 to 2.23); p=0.28 | | |
| South West | 0.86 (0.51 to 1.44); p=0.56 | 1.01 (0.60 to 1.70); p=0.97 | 0.98 (0.58 to 1.65); p=0.95 | | |
| West Midlands | 0.74 (0.44 to 1.26); p=0.26 | 0.79 (0.46 to 1.33); p=0.37 | 0.78 (0.46 to 1.31); p=0.34 | | |
| Yorkshire and The Humber | 0.87 (0.46 to 1.65); p=0.68 | 0.88 (0.47 to 1.66); p=0.70 | 0.87 (0.46 to 1.63); p=0.67 | | |
| **Drugs prior to primary KR** | | | | | |
| Oral glucocorticosteroid therapy | 0.75 (0.56 to 1.02); p=0.07 | 0.72 (0.53 to 0.99); **p=0.04** | 0.69 (0.50 to 0.94); p=0.02 | | |

**Table 3** Continued

| Risk factors (reference category) | Patients undergoing KR (n=17378) | | | Patients undergoing KR with missing dose for bisphosphonates and opioids excluded (n=14470) | |
|---|---|---|---|---|---|
| | Crude analysis | Adjusted analysis | Adjusted competing risk analysis | Adjusted analysis | Adjusted competing risk analysis |
| | | (Drug yes/no) | (Drug yes/no) | (Drug DDD) | (Drug DDD) |
| | HR (95% CI) | HR (95% CI) | HR (95% CI) | HR (95% CI) | HR (95% CI) |
| | P value | P value | P value | P value | P value |
| Drugs that can affect fracture risk prior to primary KR | | | | | |
| Antiarrhythmics | 1.35 (0.97 to 1.87); p=0.08 | 1.41 (1.00 to 1.98); **p=0.05** | 1.36 (0.97 to 1.92); p=0.078 | | |
| Anticonvulsants | 1.72 (1.11 to 2.68); **p=0.02** | 1.58 (1.01 to 2.47); **p=0.04** | 1.50 (0.96 to 2.34); p=0.076 | | |
| Painkillers/anti-inflammatory drugs | | | | | |
| Total opiates | 1.40 (1.13 to 1.73); **p<0.01** | 1.36 (1.08 to 1.71); **p=0.01** | 1.32 (1.05 to 1.65); p=0.019 | | |
| DDDs 1 year prior to primary KR | | | | | |
| Bisphosphonates (no dose) | | | | | |
| <140 DDD | 0.25 (0.03 to 1.79); p=0.17 | | | 0.40 (0.06 to 2.91); p=0.37 | 0.36 (0.05 to 2.59); p=0.31 |
| ≥140 to 340 DDD | 1.47 (0.73 to 2.96); p=0.28 | | | 2.44 (1.12 to 5.36); **p=0.03** | 2.10 (0.96 to 4.60); p=0.063 |
| >340 DDD | 0.55 (0.14 to 2.21); p=0.40 | | | 1.08 (0.26 to 4.54); p=0.92 | 0.96 (0.23 to 4.06); p=0.95 |
| Dose missing | 1.23 (0.51 to 2.95); p=0.65 | | | | |
| Opioids total (no dose) | | | | | |
| <85 DDD | 1.45 (0.95 to 2.21); p=0.09 | | | 1.33 (0.86 to 2.06); p=0.20 | 1.30 (0.84 to 2.01); p=0.25 |
| ≥85 to 365 DDD | 1.36 (0.97 to 1.90); p=0.07 | | | 1.27 (0.90 to 1.79); p=0.17 | 1.22 (0.86 to 1.72); p=0.26 |
| >365 DDD | 1.85 (1.20 to 2.85); **p=0.01** | | | 1.67 (1.08 to 2.59); **p=0.02** | 1.53 (0.99 to 2.38); p=0.056 |
| Dose missing | 1.28 (0.95 to 1.72); p=0.10 | | | | |

HR represents number of times to have a revision after 5 years compared with the reference group. A value >1 indicates that the group has higher risk for revision.

Variables included in the final regression model are those with at least one category with a p value <0.05 for the 10 imputed datasets in a backward selection.

Body mass index and sex were force-entered into all models. 'Total opiates' includes benzomorphan derivatives, diphenylpropylamine derivatives, morphinan derivatives, natural opium alkaloids, oripavine derivatives, phenylpiperidine derivatives and other opioids.

DDD, daily defined dose; KR, total and unicompartmental knee replacement.

due to including data from 2009 onwards but only primary operations up to 2011 to allow for revision rates after 5 years. This was to allow us to explore the impact of preoperative PROMs data, which has only been collected since 2009. Strengths of NJR data are detailed surgical and hospital factors available in the data. A limitation is that there have been changes in anaesthesia and surgical techniques over time that may no longer reflect current orthopaedic practice. The strength of our CPRD GOLD dataset was over 20 years of follow-up and the ability to capture a wide range of primary and hospital factors. There were missing data for some of the variables in our data, and this required us to use imputation to account for this in our analyses.

One of the aims of our study was to provide an evidence base for any group of patients in need of routine

**Table 4** Cox regression model identifying risk factors of revision after 5 years of primary total knee and unicompartmental replacement: hospital data

| Risk factors (reference category) | Patients undergoing KR (n=188 509) | | Adjusted analysis competing risks HR (95% CI); p value |
|---|---|---|---|
| | Crude analysis HR (95% CI); p value | Adjusted analysis HR (95% CI); p value | |
| Year of primary KR (2008) | | | |
| 2009 | 0.91 (0.78 to 1.06); p=0.23 | 0.90 (0.77 to 1.05); p=0.20 | 0.88 (0.75 to 1.03); p=0.10 |
| 2010 | 0.82 (0.68 to 0.98); **p=0.03** | 0.82 (0.69 to 0.99); p=0.037 | 0.77 (0.64 to 0.92); p=0.004 |
| 2011 | 0.83 (0.64 to 1.07); p=0.15 | 0.83 (0.65 to 1.07); p=0.15 | 0.69 (0.54 to 0.87); p=0.002 |
| Age at primary KR (continuous variable) | 0.94 (0.9–0.9); **p<0.01** | 0.95 (0.95 to 0.96); p<0.01 | 0.95 (0.94 to 0.95); p<0.01 |
| Sex (women) | | | |
| Men | 1.08 (1.0–1.2); p=0.23 | 1.13 (0.99 to 1.28); p=0.074 | 1.09 (0.95 to 1.24); p=0.21 |
| Body mass index (normal) | | | |
| Underweight | 1.96 (0.96 to 4.01); p=0.07 | 2.31 (1.13 to 4.73); p=0.022 | 2.22 (1.08 to 4.56); p=0.029 |
| Overweight | 1.04 (0.85 to 1.28); p=0.68 | 0.91 (0.74 to 1.11); p=0.35 | 0.92 (0.75 to 1.13); p=0.45 |
| Obese class I (moderately obese) | 1.02 (0.83 to 1.25); p=0.87 | 0.74 (0.60 to 0.91); p=0.004 | 0.75 (0.61 to 0.92); p=0.007 |
| Obese class II and higher | 1.20 (0.96 to 1.49); p=0.10 | 0.70 (0.56 to 0.88); p=0.002 | 0.71 (0.56 to 0.88); p=0.002 |
| IMD (quintiles), at primary KR (less deprived 20%) | | | |
| Less deprived 20%–40% | 0.87 (0.72 to 1.05); p=0.14 | 0.84 (0.70 to 1.01); p=0.06 | 0.84 (0.70 to 1.01); p=0.058 |
| Less deprived 40%–60% | 0.91 (0.75 to 1.10); p=0.32 | 0.78 (0.64 to 0.94); p=0.01 | 0.77 (0.64 to 0.93); p=0.008 |
| More deprived 20%–40% | 0.94 (0.78 to 1.14); p=0.55 | 0.79 (0.65 to 0.96); p=0.016 | 0.78 (0.64 to 0.94); p=0.01 |
| Most deprived 20% | 0.87 (0.71 to 1.06); p=0.17 | 0.71 (0.58 to 0.87); p=0.001 | 0.70 (0.58 to 0.86); p=0.001 |
| Ethnicity (white) | | | |
| Non-white | 0.68 (0.5 to 0.9); **p=0.01** | 0.58 (0.43 to 0.78); p<0.01 | 0.59 (0.44 to 0.80); p=0.001 |
| OKS, baseline score (0–10 points) (0=poor, 48=good) | | | |
| (11–14 points) | 0.82 (0.7 to 1.0); **p=0.03** | 0.85 (0.70 to 1.02); p=0.073 | 0.85 (0.71 to 1.02); p=0.087 |
| (15–19 points) | 0.69 (0.6 to 0.8); **p<0.01** | 0.71 (0.60 to 0.85); p<0.01 | 0.73 (0.61 to 0.87); **p<0.01** |
| (20–24 points) | 0.51 (0.4 to 0.6); **p<0.01** | 0.55 (0.44 to 0.68); p<0.01 | 0.56 (0.45 to 0.69); **p<0.01** |
| (25–48 points) | 0.37 (0.3 to 0.5); **p<0.01** | 0.42 (0.33 to 0.53); p<0.01 | 0.43 (0.34 to 0.54); **p<0.01** |
| OKS, 6-month score (0–10 points) (0=poor, 48=good) | | | |
| (11–14 points) | 0.72 (0.61 to 0.86); p<0.01 | 0.81 (0.67 to 0.96); p=0.016 | 0.81 (0.68 to 0.97); p=0.019 |
| (15–19 points) | 0.53 (0.44 to 0.63); p<0.01 | 0.59 (0.49 to 0.72): p<0.01 | 0.60 (0.50 to 0.72); p<0.01 |
| (20–24 points) | 0.43 (0.35 to 0.52); p<0.01 | 0.48 (0.39 to 0.59); p<0.01 | 0.48 (0.39 to 0.59); p<0.01 |
| (25–48 points) | 0.29 (0.23 to 0.36); p<0.01 | 0.33 (0.26 to 0.41); p<0.01 | 0.33 (0.26 to 0.42); p<0.01 |
| EQ-5D-3L Anxiety Depression, 3 months or closer to primary KR (I am not anxious or depressed) | | | |
| I am moderately anxious or depressed | 1.02 (0.9 to 1.2); p=0.78 | 0.73 (0.63 to 0.83); p<0.01 | 0.72 (0.63 to 0.82); p<0.01 |
| I am extremely anxious or depressed | 1.26 (0.9 to 1.7); p=0.14 | 0.67 (0.49 to 0.91); p=0.01 | 0.65 (0.48 to 0.89); p=0.007 |

HR represents number of times to have a revision after 5 years compared with the reference group. A value >1 indicates that the group has higher risk for revision.

Variables included in the final regression model are those with at least one category with a p value <0.05 for a single imputed dataset in a backward selection.

Body mass index and sex were force-entered into all models.

Bold figures represent results with p values <0.05 in the final regression model

EQ-5D-3L, EuroQol five domains; IMD, Index of Multiple Deprivation; KR, total and unicompartmental knee replacement; OKS, Oxford Knee Score.

follow-up after KR. Our findings were consistent with a previous study using data from the CPRD GOLD in which the authors demonstrated an instantaneous risk of revision (risk of revision following a given period of implant survival) by age and gender subgroups.[25] The smoothed hazard plots consistently showed higher revision risks for men and younger patients at all timepoints. These graphs also showed that the trends in time to revision surgery were similar across all age bands, except for the most elderly patient groups in whom follow-up is limited by life expectancy. Males and younger patients were at a consistently higher revision risk over the whole follow-up, and these factors did not influence timing of when revision occurred. In our previous work, we have shown that younger age, males and obesity are risk factors for revision hip and knee replacement.[26] Our finding in respect of age is consistent with this existing literature, as is the effect of males, showing that these effects are also seen in mid–late term, and the results were unchanged by the competing risk of death. For obesity, the opposite effect was seen in the present study where this now had a protective effect on risk of mid–late revision, although the cause of this effect was not clear. With regard to deprivation, it has previously been shown that those in the most deprived areas are less likely to receive revision knee replacement surgery,[27] and this is disappointingly consistent with what we observed and may reflect inequalities in access to revision surgery. Alternatively, it could be that obese patients or those of non-white ethnicity are more likely to be having revision surgery in the early term at less than 5 years, and hence these groups are under-represented for mid–late term revisions. However, the effect of deprivation and that of obesity were only present in the secondary care dataset, which requires further investigation.

There have been previous studies looking at the effects of medication use on revision risk, particularly for medications associated with bone and fracture risk. It has been suggested that postoperative statin use reduces revision risk for hip replacement.[28] The effects seen here in our study showed that, in crude unadjusted analyses, statins reduced the risk for knee replacement revision, but this was attenuated in the full regression model, which may be explained by the association of statin use with obesity. Bisphosphonate use has also been suggested to reduce revision risk,[29] but we saw an opposite effect for high DDD users. They had increased revision risk, which may be associated with the reason for revision as Danish studies have shown that, although bisphosphonates reduced overall all-cause revision, the risk of revision for infection was increased.[30]

The findings of this study suggest that patients receiving a mid–late revision surgery are a healthier, affluent group of patients of white ethnicity. It is unclear to what extent this represents need for revision surgery as this group may be more active, healthier, with lifestyle effects; or, is this a reflection of the known measurement error in using revision surgery as an outcome measure for the success of surgery? This patient group may simply be better able to navigate the care pathway (as for the primary operation), or reflect biases in patient–surgeon decision making, and may not be representative of those requiring revision surgery. There will always be patients in pain and functional difficulty that do not seek help from their GP or surgeon. It is of major interest to better understand why patient demographic characteristics seem to play a role in knee revision surgery.

The findings in respect of pain were interesting: in the secondary care data, although pain and function at or 6 months after primary surgery were associated with reduced risk of revision, those with the poorest scores were more likely to undergo revision. In primary care data, preoperative pain medication was the only risk factor of interest other than healthy patient case-mix selection effects that are unlikely to be informative for extended follow-up. Use of oral glucocorticoid steroid therapy may be a surrogate marker for chronic health conditions and was associated with a lower risk of revision in our data. This may reflect reduced functional goals or expectations in this patient group, with less likelihood of proceeding to revision surgery, or a reluctance to proceed with surgery due to an increased risk of infection. Anticonvulsants (gabapentin and pregabalin) and opioid use preoperatively were associated with an increased mid–late revision risk. Although opioids may be recommended for controlling pain due to osteoarthritis before primary surgery,[31] they may also be indicative of chronic pain and/or opioid related comorbidities, and two-thirds of patients have been shown to continue to use opioids postsurgery.[29 32] This group of patients often experience a mixed picture of pain and may have high levels of dissatisfaction after surgery, leading them to seek further surgical solutions for persistent pain.[33 34] Use of anticonvulsants prior to primary surgery is suggestive of existing neuropathic pain or multisite joint pain. Postsurgery, this group of patients may experience sensitisation subsequent to chronic pain and/or additional neuropathic components, leading to more severe symptoms that places them at greater risk of revision. Further work would be required to investigate whether patients with neuropathic or chronic pain after primary KR would benefit from closer monitoring and follow-up, particularly if they are then at further increased risk of mid–late revision.

In this study, there was an opportunity to examine unique datasets for predictors of mid–late term revision risk for KR surgery. We have reported the results for KR in this study, but it is of interest that in the wider programme of work (UK SAFE[12]), the predictors of revision were different for hips and knees with age being the main consistent finding. The patient factors we identified as predictive of mid-late term revision risk after KR may reflect inequalities in access to revision surgery, or there may be other factors not captured within this study; this requires further investigation. In addition, further work is needed to determine if targeted follow-up is required for those patients with worse pain and function preprimary

and/or postprimary surgery, or higher levels of preoperative pain medication (opioids and anticonvulsants) due to their increased risk of mid–late term revision. The findings from this study have implications for future provision of follow-up services for patients with a KR.

**Author affiliations**
[1]Department of Health & Applied Sciences, University of the West of England, Bristol, UK
[2]Nuffield Department of Orthopaedics Rheumatology and Musculoskeletal Sciences, University of Oxford, Oxford, UK
[3]NIHR Leeds Biomedical Research Centre, Leeds, UK
[4]Leeds Institute of Rheumatic and Musculoskeletal Medicine, University of Leeds, Leeds, UK
[5]Nuffield Department of Orthopaedics, Rheumatology and Musculoskeletal Sciences, University of Oxford, Oxford, UK
[6]MRC Lifecourse Epidemiology Unit, University of Southampton, Southampton, UK
[7]Orthopaedic Department, Leeds Teaching Hospitals NHS Trust, Leeds, UK
[8]Translational Health Sciences, Bristol Medical School, University of Bristol, Bristol, UK

**Acknowledgements** We gratefully acknowledge interpretation offered by Mr Nicholas Howells (North Bristol NHS Trust). We would like to thank the patients and staff of all the hospitals in England and Wales who have contributed data to the National Joint Registry (NJR); and the Healthcare Quality Improvement Partnership, the NJR Steering Committee and staff at the NJR Centre for facilitating this work. The authors have conformed to the NJR's standard protocol for data access and publication.

**Contributors** LKS, SRK, NKA, MS, PGC and AJ in conception of work, approval of final version and accountability. AJ acts as guarantor. CG, AD and AJ: analysis of data; AJ, SRK, MS, LKS, RP-V and PGC: interpretation of results; AJ and LKS drafted and revised final manuscript.

**Funding** This article presents independent research funded by the National Institute for Health Research (NIHR) Health Services and Delivery Research Programme (14/70/146) and by the NIHR Leeds Biomedical Research Centre (BRC). SRK, MS and PGC were supported in part by the NIHR Leeds Biomedical Research Centre. AJ was supported by the NIHR Biomedical Research Centre at University Hospitals Bristol and Weston NHS Foundation Trust and the University of Bristol.

**Competing interests** LS reports grants from NIHR during the conduct of the study. AJ reports grants from NIHR and has received consultancy fees from Freshfields Bruckhaus Derringer and Anthera Pharmaceuticals LTD unrelated to this work. AJ was supported by the NIHR Biomedical Research Centre at University Hospitals Bristol and Weston NHS Foundation Trust and the University of Bristol. NA reports grants from Merck, personal fees from Pfizer/Lilly, unrelated to the submitted work. SRK, MS and PGC were supported in part by the NIHR Leeds Biomedical Research Centre.

**Patient consent for publication** Not applicable.

**Ethics approval** The CPRD Group has obtained ethical approval from a National Research Ethics Service Committee (NRES) for all purely observational research using anonymised CPRD data; namely, studies that do not include patient involvement. The study has been approved by Independent Scientific Advisory Committee for MHRA Database Research (protocol number 11_050AMnA2RA2). For the NJR, before personal data and sensitive personal data are recorded, express written patient consent is provided. The NJR records patient consent as either 'Yes', 'No' or 'Not Recorded'. With support under Section 251 of the NHS Act 2006, the Ethics and Confidentiality Committee (now the Health Research Authority Confidentiality Advisory Group) allows the NJR to collect patient data where consent is indicated as 'Not Recorded'. Section 251 support (17/CAG/0030) and NHS Digital approval (DARS-NIC-172121-G0Z1H-v0.11) were obtained for the NJR-HES-PROMS dataset analysis.

**Provenance and peer review** Not commissioned; externally peer reviewed.

**Data availability statement** Data may be obtained from a third party and are not publicly available. Access to data is available from the National Joint Registry for England and Wales, Northern Ireland and the Isle of Man, but restrictions apply to the availability of these data, which were used under license for the current study, and so are not publicly available. Data access applications can be made to the National Joint Registry Research Committee. Access to linked Hospital Episode Statistics and PROMs data is available through data applications to NHS Digital.

**ORCID iDs**
Lindsay K Smith http://orcid.org/0000-0002-9979-3180
Philip G Conaghan http://orcid.org/0000-0002-3478-5665

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
