## [Reviewer comments · BMJ Open]

ARTICLE DETAILS

TITLE (PROVISIONAL)	UK poSt Arthroplasty Follow-up rEcommendations (UK SAFE): What does analysis of linked, routinely collected national datasets tell us about mid-late term revision risk after knee replacement?
AUTHORS	Smith, Lindsay; Garriga, Cesar; Kingsbury, Sarah; Pinedo-Villanueva, Rafael; Delmestri, Antonella; Arden, Nigel; Stone, Martin; Conaghan, Philip; Judge, Andrew

VERSION 1 – REVIEW

REVIEWER	EI-Galaly, Anders Aalborg University Hospital, Department of Orthopedic Surgery
REVIEW RETURNED	07-Feb-2021

GENERAL COMMENTS	Thank you for the opportunity to review this interesting paper and I commend the authors on their work in relation to it. The study aims to identify risk factors associated with the mid/long-term revisions of knee replacement (KR), defined as revision ≥ 5 years following primary KR. The study is based on two large (intersecting?) nationwide British databases. I find the research question important and interesting, and the manuscript well written. However, I suspect that the study has some major methodological limitations. Furthermore, the study, as presented, does not add substantial new knowledge to the current literature. Yet, if the methodological issues are addressed and the 6-months postoperative PROMs are included, I think the study will be suitable for publication in the BMJ Open. Below are my major concerns regarding this study, which I think should be addressed before the study can be reconsidered for publication. I will gladly provide a more profound review of an updated version of this study. First, the methodological issues: 1. I strongly recommend the authors to report the study in accordance with the STROBE guidelines and state that they have done so. Not only will it increase the quality of the manuscript, but it will also make the study easier to include in future reviews, meta-analyses etc.2. The study is based on two datasets from two nationwide databases within the same country. As the datasets cover the same period, I assume that some patients, if not all, are included in both datasets. This is a major methodological issue, as the relative few revised KRs will be present in both datasets. This give the false
---

impression that the presented risk factors are consistent across independent datasets.
If there is no overlap between the datasets, the authors should state this clearly.
Yet, if there is a major overlap, the authors should address this in a revised manuscript. I have the following suggestions on how to address this concern:
If the authors want to present two datasets, I think it is a prerequisite for publication that they analyze the overlap of KRs between the datasets. One approach is to exclude any KRs which is already included in the other dataset. Or at least, state clearly how large the overlap is and how it limits the results of the study.
A more valued approach is to merge the two datasets, and thereby presenting only one dataset with a single observation for each KR containing information from both datasets.
If all records are anonymized and linkage between the two datasets is impossible, I recommend the authors to include only one of the datasets in order to avoid presenting results from the same KR twice.

3. The study does not include any competing risk analyses when estimating the association between “predictors” and KR revision. Especially when analyzing longer follow-up in an older population, the risk of dying before revision might bias the results. I have added a few references, which might be helpful for the authors.
I might suspect that the finding that non-white, obesity, living in deprived areas and older age protects against long-term revision might be biased by death occurring before revision is more frequent in these groups. I suggest that the authors include a competing risk regression, such as Fine-Gray’s, as a sensitivity analysis to address this concern. Competing risk analyses might be more appropriate in this study, given that the ultimate goal is to define the ideal follow-up of patients receiving knee replacement and not the survival of the individual implants.

Gillam MH, Ryan P, Graves SE, Miller LN, De Steiger RN, Salter A. Competing risks survival analysis applied to data from the Australian orthopaedic association national joint replacement registry. *Acta Orthop.* 2010;81:548–555.

Sayers A, Evans JT, Whitehouse MR, Blom AW. Are competing risks models appropriate to describe implant failure? *Acta Orthop.* 2018;89:256–258.

Secondly, as mentioned, I do not think that the study provides substantially new knowledge to the current literature. However, the authors states that 6-months post-operative PROMs are available from KRs in the NJR cohort. Yet, they do not analyze if these are correlated with the risk of mid/long-term revision. To my knowledge, the influence of post-operative PROMs on the risk of long-term knee placement revision is seldom investigated. Often, because many registries do not collect PROMs at follow up.
I encourage the authors to investigate the possible association between 6-month postoperative PROMs and long-term risk of revision. I suspect that early post-operative measures, especially PROMs, have a strong correlation to mid/long-term revision. Maybe the individualized follow-up protocol after KR should be determined based on the 6-months PROMs and patient demographics?

	Exploring this aspect would be a major improvement of this study and, in my opinion, provide substantially new knowledge to the field of arthroplasty research.
--	---

REVIEWER	Bass, Anne R Hospital for Special Surgery
REVIEW RETURNED	31-Mar-2021

GENERAL COMMENTS	This study looked at risk factors for what they term mid-late term knee replacement revision surgery using the Clinical Practice Research Datalink 1997-2017 and National Joint Registry dataset 2009-2017, as well as a dataset with PROMs. Only TKR performed before 2012 were included so that all patient had at least five years of follow up, and revisions occurring before 5 years were excluded. The authors state they are motivated by recent guidelines that did not explicitly recommend follow up of TKR patients long term due to lack of evidence for a benefit. They argue that early detection of failing implants and targeted follow up should be performed (I'm not necessarily convinced) and that identifying risk factors for late revision will allow such targeting. The CPRD GOLD-HES dataset had a longer time-to-revision. There were 877 (5.1%) revisions, with median time-to-revision of 4.2 years (range 0.02 to 18.3 years) and 352 (2.0%) were mid-late term revision. In the NJR-HES-PROMs data, there were 8,607 (4.6%) knee replacement revisions with median time-to-revision of 1.8 years (range 0 to 8.8 years); this included 1,055 (0.6%) mid-late term revisions. They found, as in other studies, that older age, female sex and obesity were associated with a lower risk of mid-late revision. Better functional status before primary knee replacement was associated with a higher rate of mid-late TKR. The major issue with this study is the exclusion of patients who underwent revisions before five years, which is not adequately justified by the authors, and which makes interpretation of the data difficult. They in fact refer to an earlier study (reference 22 Bayliss Lancet 2017) demonstrating "smooth hazard plots...showing higher revision risks for men and younger patients at all time-points". So why arbitrarily eliminate the revisions prior to 5 years? A better approach would have been to perform a landmark analysis, or several landmark analyses with the results compared for different time cutoffs. The authors should also perform sensitivity analyses comparing the >50% of cases excluded from their analysis (e.g. due to lack of linkage to the other datasets) to those included to make sure the sample is representative. The authors excluded non-OA diagnoses, thereby missing an opportunity to analyze inflammatory arthritis as a potential risk factor for revision. Finally, in their discussion the authors never return to the supposed premise of their study, to guide "targeted follow up". What do they recommend be done policy wise? How should their data be used?
--

REVIEWER	Kuklinski, David Technische Universität Berlin, Health Care Management
REVIEW RETURNED	07-Apr-2021

GENERAL COMMENTS	(1) The topic of optimal aftercare for patients after KR is an important topic to steer efforts and resources efficiently. I like the topic, and also how the authors address the question of which patient groups are especially affected by revision risks. (2) Introduction: The authors span a good background story to the existing research gap of investigating optimal postoperative follow-up period and the best long-term care model for people with OA and knee replacement. Unfortunately, the objective of the paper does not answer / address this research gap. Also the results rather point to patient characteristics that increase the risk for revisions, and not so much to the optimal follow-up period for knee replacement patients. It would be great to better bridge between background and research question. (3) Methods: The methods are clearly laid out and possible biases are addressed. The authors explain the data sources and data matching process in detail. (4) Data: Is it possible to link both datasets? This would be a possibility to include all possible patient data into the model. Why are they analyzed separately? (4) Results: The results are quite counter-intuitive (e.g., younger non-obese people of white ethnicity, better PROM results at admission). What could be the reason? Could it be that older/ obese patients with worse pre-operative PROM scores are having more often surgery before the 5 years mark? And thus, these groups are underrepresented for mid-late revisions? Have you accounted for this fact? (5) Conclusion: I am missing a clear implication / conclusion following the results. What do the results imply for ideal post-operative follow-up period and the best long-term care model for people with OA and knee replacement? Why are the results important? (6) Language: The grammar, language and spelling is excellent.
--

VERSION 1 – AUTHOR RESPONSE

Reviewer: 1

Dr. Anders El-Galaly, Aalborg University Hospital

Comments to the Author:

Comment	Action	Response
1. I strongly recommend the authors to report the study in accordance with the STROBE guidelines and state that they have done so. Not only will it increase the quality of the manuscript, but it will also make the study easier to include in future reviews, meta-analyses etc	STROBE checklist included as supplementary file III.	Thank you for this suggestion.
2. The study is based on two datasets from two nationwide databases within the same country. As	We have made it clearer in the methods section, that	We have analysed two separate anonymised datasets for this paper.

the datasets cover the same period, I assume that some patients, if not all, are included in both datasets. This is a major methodological issue, as the relative few revised KRs will be present in both datasets. This give the false impression that the presented risk factors are consistent across independent datasets. If there is no overlap between the datasets, the authors should state this clearly. Yet, if there is a major overlap, the authors should address this in a revised manuscript. I have the following suggestions on how to address this concern: If the authors want to present two datasets, I think it is a prerequisite for publication that they analyze the overlap of KRs between the datasets. One approach is to exclude any KRs which is already included in the other dataset. Or at least, state clearly how large the overlap is and how it limits the results of the study. A more valued approach is to merge the two datasets, and thereby presenting only one dataset with a single observation for each KR containing information from both datasets. If all records are anonymized and linkage between the two datasets is impossible, I recommend the authors to include only one of the datasets in order to avoid presenting results from the same KR twice.	the CPRD data contains a representative sample of 6.9% of the population. We further state in the methods that there will be some overlap between patients receiving knee replacement in the two data sources (around 7% of patients between 2009 and 2016), however these anonymised datasets are analysed independently of each other.	The first is a secondary care dataset from the National Joint Registry that has been linked to English Hospital Episode Statistics (HES) and Patient Reported Outcome Measures (PROMS) from 2009 to 2017. This national dataset captures all patients receiving knee replacement surgery. The second is data from primary care (Clinical Practice Research Datalink (CPRD)) linked to HES hospital records for a period from 1995 to 2016. CPRD is a sample of data of approximately 6.9% of the UK population and patients are broadly representative of the UK general population in terms of age, sex and ethnicity. The overlap between the two datasets would therefore be around 7% between the years 2009 to 2016 (with no overlap for earlier years of CPRD data), hence there is no major overlap, and the datasets are analysed independently from one another. Both datasets are anonymised and cannot be linked together. The advantage of the NJR-HES-PROMS data is that we can explore surgical and operative factors, and data on PROMS, that are not captured within the other dataset. The advantage of the CPRD dataset, is that we can explore data such as medication use, which is not available within the NJR. Hence, we feel that these analyses are complimentary and provide valuable additional information from including both data sources. Not including the NJR data would mean we could not include analyses of PROMS as requested by reviewers; not including CPRD data would mean information about medication use is not presented. However, if the editors feel strongly that only one data source should be included, we would be happy to do so.
3. The study does not include any competing risk analyses when	We have conducted sensitivity analyses,	We have conducted sensitivity analyses, using Fine-Gray models

estimating the association between “predictors” and KR revision. Especially when analyzing longer follow-up in an older population, the risk of dying before revision might bias the results. I have added a few references, which might be helpful for the authors. I might suspect that the finding that non-white, obesity, living in deprived areas and older age protects against long-term revision might be biased by death occurring before revision is more frequent in these groups. I suggest that the authors include a competing risk regression, such as Fine-Gray’s, as a sensitivity analysis to address this concern. Competing risk analyses might be more appropriate in this study, given that the ultimate goal is to define the ideal follow-up of patients receiving knee replacement and not the survival of the individual implants. Gillam MH, Ryan P, Graves SE, Miller LN, De Steiger RN, Salter A. Competing risks survival analysis applied to data from the Australian orthopaedic association national joint replacement registry. Acta Orthop. 2010;81:548–555. Sayers A, Evans JT, Whitehouse MR, Blom AW. Are competing risks models appropriate to describe implant failure? Acta Orthop. 2018;89:256–258.	using Fine-Gray models to account for the competing risk of death.	to account for the competing risk of death. The results are very similar, and findings and conclusions remain unchanged.
To my knowledge, the influence of post-operative PROMs on the risk of long-term knee placement revision is seldom investigated. Often, because many registries do not collect PROMs at follow up. I encourage the authors to investigate the possible association between 6-month postoperative PROMs and long-term risk of revision. I suspect that early post-operative measures, especially PROMs, have a strong correlation to mid/long-term revision. Maybe the individualized follow-up protocol after KR should be determined based on the 6-months PROMs and patient demographics? Exploring this aspect would be a major improvement of this study and, in my opinion, provide substantially new knowledge to the field of arthroplasty research.	The 6-month OKS is now included in the models.	For the NJR-HES-PROMS dataset, we have now included the 6-month PROMs score in the regression model. The reviewer was correct that this is indeed associated with the risk of mid-long term revision.

Reviewer: 2

Dr. Anne R Bass, Hospital for Special Surgery

Comments to the Author:

Comment	Action	Response
The major issue with this study is the exclusion of patients who underwent revisions before five years, which is not adequately justified by the authors, and which makes interpretation of the data difficult. They in fact refer to an earlier study (reference 22 Bayliss Lancet 2017) demonstrating “smooth hazard plots...showing higher revision risks for men and younger patients at all time-points”. So why arbitrarily eliminate the revisions prior to 5 years? A better approach would have been to perform a landmark analysis, or several landmark analyses with the results compared for different time cutoffs.	We have made clearer in the methods section, the rationale for determining mid-late term revision as being defined as occurring after 5-years.	This study has been conducted for an NIHR Health Services & Delivery Research grant, where the planned analyses have already undergone a process of peer review. Our a-priori research question was to identify predictors of mid-late term revision, with these being defined as occurring after 5-years. The rationale for follow-up is to ensure timely detection of complications or arthroplasty failure that will require revision surgery. Early complications (defined as less than five years) are often symptomatic and include infection and technical errors [Reference 21]. Arthroplasty failure in the longer term (defined as after five years), constituting 50% of revision surgery, is usually caused by bearing-surface wear and associated consequences of periprosthetic osteolysis or aseptic loosening and may be asymptomatic until clinical and radiographic failure have occurred [References 21,22]. Hence, our rationale for focusing on mid-late term revision.
The authors should also perform sensitivity analyses comparing the >50% of cases excluded from their analysis (e.g. due to lack of linkage to the other datasets) to those included to make sure the sample is representative. The authors excluded non-OA diagnoses, thereby missing an opportunity to analyze inflammatory arthritis as a potential risk factor for revision	We have included in an appendix a table of descriptive statistics, comparing the characteristics of patients in the CPRD dataset, versus those in the CPRD-HES subset with linked data.	The dataset the reviewer refers to is specifically the CPRD-HES linked dataset, where HES data can only be linked for those GP practices that take part in the linkage scheme. We have therefore provided descriptive statistics, comparing the characteristics of patients in the CPRD dataset, versus those in the CPRD-HES subset with linked data. Our research question is for those patients receiving primary knee replacement for osteoarthritis. Whilst we agree with the reviewer that it would be of interest to look at patients with knee replacement for inflammatory arthritis, this is beyond the scope of the current study.
Finally, in their discussion the authors never return to the supposed premise of their study, to guide “targeted follow up”. What do they recommend be done policy wise? How should their data be used?		We appreciate the reviewer’s comments but would respectfully point out that this study was one part of a programme of work (UK SAFE) on this topic. This particular study was intended to examine existing large datasets to identify

		any groups of patients at higher risk for mid-late term revision. These results have been integrated with the results from the other studies to produce the final report, which is due for publication imminently, and which will be available for future policy makers to consider. It is our hope that the final report will inform future policy and future research.
--	--	--

Reviewer: 3

Dr. David Kuklinski, Technische Universität Berlin

Comments to the Author:

Comment	Action	Response
(1) The topic of optimal aftercare for patients after KR is an important topic to steer efforts and resources efficiently. I like the topic, and also how the authors address the question of which patient groups are especially affected by revision risks.	None	We thank Reviewer 3 for his comments on the manuscript and his affirmation of the topic.
(2) Introduction: The authors span a good background story to the existing research gap of investigating optimal postoperative follow-up period and the best long-term care model for people wit OA and knee replacement. Unfortunately, the objective of the paper does not answer / address this research gap. Also the results rather point to patient characteristics that increase the risk for revisions, and not so much to the optimal follow-up period for knee replacement patients. It would be great to better bridge between background and research question.		We agree with the reviewer with respect to the gap in the literature, hence the programme of work (UK SAFE) of which this study was one part. Our objective in this study was specifically to identify groups of patients who may be at higher risk for mid-late term revision. We believe that we fulfilled that objective. The intention was that other parts of the UK SAFE programme would provide results from which we could make informed statements about the postoperative follow-up period and long-term care models. The report of the programme of work is due for publication imminently, and to date, there is one paper in press awaiting publication, and others under review.
(4) Data: Is it possible to link both datasets? This would be a possibility to include all possible patient data into the model. Why are they analyzed separately?	Please see earlier response to reviewer 1.	Please see earlier response to reviewer 1. These are separate anonymised datasets and cannot be linked.
(4) Results: The results are quite counter-intuitive (e.g., younger not-	We have expanded the discission to	Our research question is to identify predictors of mid-late term revision.

obese people of white ethnicity, better PROM results at admission). What could be the reason? Could it be that older/ obese patients with worse pre-operative PROM scores are having more often surgery before the 5 years mark? And thus, these groups are underrepresented for mid-late revisions? Have you accounted for this fact?	incorporate the interpretations suggested by the reviewer.	Our current interpretation of these findings in the discussion section, is that they are likely to reflect inequalities in access to revision surgery, rather than a true increased risk of revision. An alternative explanation, as the reviewer suggests is that these groups have increased revision risk in the early term, which would be consistent with existing literature on risk factors such as age and obesity, hence we have expanded on this in the discussion. In our separate paper submitted to BMJ Open for hip replacement, we do not see any associations for age, obesity, white ethnicity and PROM scores, and so it seems these findings are specific to knee replacement.
(5) Conclusion: I am missing a clear implication / conclusion following the results. What do the results imply for ideal post-operative follow-up period and the best long-term care model for people with OA and knee replacement? Why are the results important?		We understand the questions raised by the reviewer, but this study was not designed to answer these questions in a standalone fashion – it is just one part of a programme of work (UK SAFE) on this topic. This particular study was intended to examine existing large datasets to identify any groups of patients at higher risk for mid-late term revision. The results imply that patients with certain characteristics are at higher risk for mid-late term revision, which requires further work to understand why this is the case. The results also showed that there is a higher risk for revision for those patients who experience worse pain or function before or shortly after primary surgery, or who require particular types of pain relief prior to primary surgery. This also requires further work to understand if they would benefit from closer monitoring or follow-up. These results are integrated with the results from the other studies to produce the final report which is due for publication imminently.

VERSION 2 – REVIEW

REVIEWER	Bass, Anne R Hospital for Special Surgery
REVIEW RETURNED	14-Oct-2021

GENERAL COMMENTS

The authors have made few changes in response to reviewer suggestions, particularly with regard to bias introduced by eliminating revisions occurring in the first five years. The authors performed a competing risk analysis using death as the competing risk, but revision prior to 5 years is the true competing risk (as demonstrated by the paradoxical finding of increased revision risk after 5 years in white affluent individuals, which may reflect survival bias). For many requested changes they argue that they cannot change their study design based on the program of work defined by UK SAFE. That may be the case, but it negatively impacts the significance/impact of this study.

The results demonstrate that revisions continue to occur after five years (which is known) and risk factors are so general (male sex, younger age, non-obese) that it is difficult to see how this study will inform public policy. Notably, most risk factors are the same as those in the first five years shown in other studies, suggesting that the cutoff of five years is not biologically based. The authors argue that late revisions will be asymptomatic and the patients will not present to doctors without follow up, but that is not demonstrated by their data in CPRD GOLD in which median time to revision was 4 years